# STAR: Learning Diverse Robot Skill Abstractions through Rotation-Augmented Vector Quantization

**Hao Li**[1 2]  **Qi Lv**[1 2]  **Rui Shao**[† 1]  **Xiang Deng**[† 1]  **Yinchuan Li**[2]  **Jianye Hao**[2]  **Liqiang Nie**[1]

STAR.github.io

## Abstract

Transforming complex actions into discrete skill abstractions has demonstrated strong potential for robotic manipulation. Existing approaches mainly leverage latent variable models, e.g., VQ-VAE, to learn skill abstractions through learned vectors (codebooks), while they suffer from codebook collapse and modeling the causal relationship between learned skills. To address these limitations, we present **S**kill **T**raining with **A**ugmented **R**otation (**STAR**), a framework that advances both skill learning and composition to complete complex behaviors. Specifically, to prevent codebook collapse, we devise rotation-augmented residual skill quantization (RaRSQ). It encodes relative angles between encoder outputs into the gradient flow by rotation-based gradient mechanism. Points within the same skill code are forced to be either pushed apart or pulled closer together depending on gradient directions. Further, to capture the causal relationship between skills, we present causal skill transformer (CST) which explicitly models dependencies between skill representations through an autoregressive mechanism for coherent action generation. Extensive experiments demonstrate the superiority of STAR on both LIBERO benchmark and realworld tasks, with around 12% improvement over the baselines.

## 1. Introduction

The challenge of modeling multitask visuomotor policy has long been a central problem in robotic manipulation (Levine

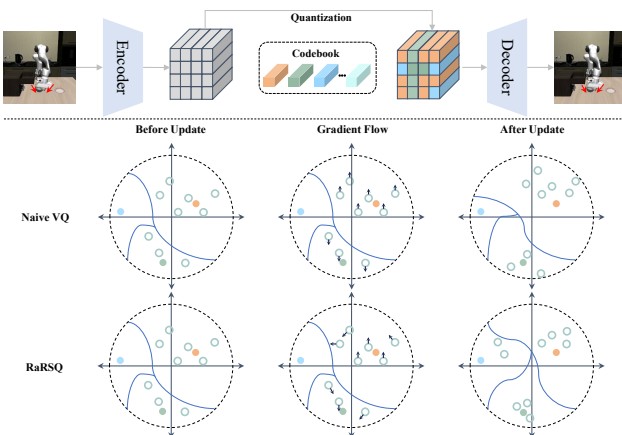

*Figure 1.* Comparison between naive VQ and our RaRSQ approach in the skill learning process. Top: Overview of skill quantization framework. Bottom: Visualization of gradient flow and codebook updates across three stages (before update, during gradient flow, and after update), where RaRSQ maintains geometric relationships between embeddings, leading to more diverse skills.

et al., 2016; Zhu et al., 2018). Individual manipulation tasks already pose significant challenges like multimodal action distributions (Mandlekar et al., 2021), while these challenges are substantially amplified in the multitask setting. This results in a highly entangled action space where characteristics of different tasks interact and overlap, making it challenging to learn complex manipulation behaviors (Lv et al., 2024).

An intuitive approach to alleviate this problem is to learn structured representations of manipulation behaviors by decomposing complex actions into simpler, reusable skill abstractions (Fu et al., 2024; Sharma et al., 2019). This hierarchical framework reflects the compositional structure inherent to manipulation tasks, enabling the systematic decomposition of complex behaviors into sequences of skill abstractions. Recent studies (Ju et al., 2024; Wu et al., 2024) have demonstrated promising results using latent variable models (LVM) to discretize continuous action spaces into learned skills. These methods enable a more efficient representation and composition of complex behaviors.

[†]Corresponding Author  [1]School of Computer Science and Technology, Harbin Institute of Technology (Shenzhen) [2]Huawei Noah's Ark Lab. Correspondence to: Rui Shao <shaorui@hit.edu.cn>, Xiang Deng <dengxiang@hit.edu.cn>.

*Proceedings of the 42ⁿᵈ International Conference on Machine Learning*, Vancouver, Canada. PMLR 267, 2025. Copyright 2025 by the author(s).

However, while this discretization paradigm transforms continuous actions into skills and provides a structured representation for complex behaviors, existing LVM-based methods face two critical limitations in skill learning and composition (Garg et al., 2022; Lee et al., 2024). First, techniques like VQ-VAE suffer from codebook collapse (Mentzer et al., 2023; Roy et al., 2018). Most codebook vectors remain unused during training, with only a small subset being frequently utilized for encoding diverse manipulation skills. This severely limits the capacity to capture the rich variety of robot behaviors. We argue that this limitation stems from the straight-through gradient estimator (STE) in VQ-VAE (Van Den Oord et al., 2017). During training, as illustrated in the middle column of Fig. 1, STE assigns identical gradients to all encoder embeddings (hollow circles) that are quantized to the same codebook vector (orange solid circles). This oversimplified gradient assignment ignores the inherent geometric relationships between different embeddings within the same partition (regions separated by curved blue decision boundaries), leading to suboptimal codebook updates and eventual collapse of the representation space.

Second, existing approaches struggle with effective skill composition, particularly in complex, long-horizon tasks that require precise coordination of multiple skills (Mete et al., 2024). While some methods adopt residual quantization (RQ) (Zeghidour et al., 2021) to decompose skills into multiple levels for more precise representation, they fail to model the dependencies between different skill abstractions (Lee et al., 2024). This makes it difficult to generate coherent and temporally consistent actions for multi-stage manipulation tasks, where skills need to be carefully sequenced and composed.

To address these fundamental challenges, we propose **S**kill **T**raining with **A**ugmented **R**otation (**STAR**), a novel framework that advances both skill learning and composition for robot manipulation. Our key insight is that encoding geometric relationships between action sequences into the residual quantization process is crucial for learning diverse and reusable skills, rather than relying on the oversimplified gradient assignment of straight-through estimation. Specifically, **(1) to prevent codebook collapse**, we devise rotation-augmented residual skill quantization (RaRSQ), which combines multi-level residual encoding with rotation-based gradient mechanisms. The residual structure progressively captures skills at different abstraction levels, while the rotation-augmented gradient flow enables points within the same skill code to be either pushed apart or pulled closer together based on their geometric relationships. Compared to naive VQ-VAE where similar embeddings are forced to have identical gradients, Fig. 1 demonstrates that RaRSQ prevents embeddings from collapsing to the same codebook vector, leading to more diverse skill representations; and **(2) for effective skill composition**, we present causal skill transformer

(CST) which explicitly models dependencies between skill representations through an autoregressive mechanism for coherent action generation. By leveraging the hierarchical nature of residual quantization, CST sequentially predicts skill codes from coarse to fine levels and incorporates an offset prediction mechanism from BeT (Shafiullah et al., 2022) to bridge the gap between discrete skills and continuous control. This combination of hierarchical prediction and continuous refinement enables precise control throughout extended sequences, making it particularly effective for long-horizon manipulation tasks. To summarize, our main contributions are as follows:

- A rotation-augmented residual skill quantization (RaRSQ) mechanism that maintains diverse skill representations by encoding relative angular relationships in gradient updates, while achieving precise skill abstraction through hierarchical residual encoding.

- A causal skill transformer (CST) that models skill dependencies through autoregressive prediction and enhances action precision via action refinement.

- Comprehensive experimental validation across multiple benchmarks and real-world tasks, demonstrating substantial improvements in both skill learning efficiency and task performance.

## 2. Related Work

**Multi-task Imitation Learning.** Multi-task robotic learning has been approached through various methods including supervised pre-training (Sun et al., 2023; Wu et al., 2023; Li et al.) and large-scale demonstration learning (Vuong et al., 2023; Brohan et al., 2023; Li et al., 2025). Recent advances have explored the use of generative models, with frameworks like diffusion models (Chi et al., 2023; Ze et al., 2024; Lv et al., 2025) and transformer-based approaches (Pertsch et al., 2025; Bharadhwaj et al., 2024) showing promising results in handling multimodal action distributions. The Behavior Transformer (BeT) (Shafiullah et al., 2022) demonstrated that policies operating in discretized action spaces can effectively model diverse behaviors, and introduced an offset prediction mechanism to handle the discretization-induced precision loss. Action Chunking Transformer (ACT) (Zhao et al., 2023) further addressed temporal correlations by predicting action chunks. Unlike existing methods, STAR advances this line of work by introducing novel mechanisms for learning structured skill representations while preserving the geometric relationships inherent in continuous action sequences.

**Robotic Manipulation in Learned Latent Spaces.** Latent Variable Models (LVMs) have emerged as powerful tools for learning structured representations in robotics (Yang

et al., 2024; Luo et al., 2023; Bharadhwaj et al., 2023), particularly for offline imitation learning and skill abstractions. Among them, methods like LAPA (Ye et al., 2024), IGOR (Chen et al., 2024), leverage internet-scale human videos to learn transferable manipulation skills. Several works have explored discrete latent spaces for skill representation. PRISE (Zheng et al., 2024) employs BPE tokenization for temporal abstraction but struggles to effectively encode varied action distributions across tasks. TAP (Jiang et al., 2022) and H-GAP (Jiang et al., 2023) use self-supervised autoencoders for skill learning but rely heavily on state-based model predictive control, limiting their real-world applicability. QuEST (Mete et al., 2024) learns discrete latent skills with temporal dependencies, but struggles to learn diverse skill representations. VQ-BeT (Lee et al., 2024) shares our motivation of using discrete latent skills for transformer-based policies, but their standard quantization approach suffers from codebook collapse and fails to capture temporal dependencies between skills. Unlike existing approaches, STAR addresses both the representational and temporal challenges through a two-stages framework that combines rotation-augmented quantization for preventing codebook collapse with explicit modeling of skill dependencies for coherent behavior generation.

## 3. Method

### 3.1. Preliminaries

**Residual VQ-VAE and STE.** VQ-VAE with residual quantization transforms continuous data into hierarchical discrete representations through a multi-stage quantization process (Adiban et al., 2022). It consists of three key components: an encoder $\mathcal{E}$, a decoder $\mathcal{D}$, and multiple codebooks $\mathcal{C}_i$. Given an input $\mathbf{x} \in \mathbb{R}^n$, the encoder $\mathcal{E}$ first maps it to a continuous latent code $\mathbf{e} \in \mathbb{R}^m$. Then, $\mathbf{e}$ are quantized using $D$ codebooks, where each codebook $\mathcal{C}_i = \{\mathbf{e}_{(i,1)}, ..., \mathbf{e}_{(i,K)}\}$ contains $K$ learnable vectors.

Starting with the initial residual $\mathbf{r}_0 = \mathbf{e}$, residual quantization iteratively performs nearest neighbor lookup and residual computation:

$$k_d = Q(\mathbf{r}_{d-1}; \mathcal{C}_d) = \arg\min_{k \in \{1,...,K\}} \|\mathbf{r}_{d-1} - \mathbf{e}_{(d,k)}\|_2^2 \tag{1}$$

$$\mathbf{r}_d = \mathbf{r}_{d-1} - \mathbf{e}_{(d,k_d)} \tag{2}$$

where $k_d$ is the selected code index at depth $d$, and $\mathbf{r}_d$ is the remaining residual to be quantized by subsequent codebooks. The final quantized representation $\hat{\mathbf{e}}$ is obtained by summing the selected code vectors:

$$\hat{\mathbf{e}} = \sum\nolimits_{d=1}^{D} \mathbf{e}_{(d,k_d)} \tag{3}$$

The decoder $\mathcal{D}$ then reconstructs the input: $\hat{\mathbf{x}} = \mathcal{D}(\hat{\mathbf{e}})$. The model is trained with a combination of reconstruction and

---

**Algorithm 1** Rotation-augmented Residual Skill Quantization (RaRSQ)

---

**Require:** action sequence $\mathbf{a}_{t:t+T}$, codebooks $\{\mathcal{C}_d\}_{d=1}^{D}$
1: $\mathbf{z} \leftarrow \text{Encoder}(\mathbf{a}_{t:t+T})$
2: $\mathbf{r}_0 \leftarrow \mathbf{z}$
3: **for** $d = 1$ **to** $D$ **do**
4:     // Quantize current residual
5:     $k_d \leftarrow \arg\min_k \|\mathbf{r}_{d-1} - \mathbf{e}_{(d,k)}\|_2^2$
6:     // Compute rotation matrix that aligns $\mathbf{r}_{d-1}$ to $\mathbf{e}_{(d,k_d)}$
7:     $\mathbf{R}_d \leftarrow \text{ComputeRotation}(\mathbf{r}_{d-1}, \mathbf{e}_{d,k_d})$
8:     // Apply rotation with stop-gradient to preserve geometric structure
9:     $\tilde{\mathbf{q}}_d \leftarrow \text{sg}\left[\frac{\|\mathbf{e}_{(d,k_d)}\|}{\|\mathbf{r}_{d-1}\|}\mathbf{R}_d\right]\mathbf{r}_{d-1}$
10:     // Update residual for next level
11:     $\mathbf{r}_d \leftarrow \mathbf{r}_{d-1} - \tilde{\mathbf{q}}_d$
12: **end for**
13: $\hat{\mathbf{z}} \leftarrow \sum_{d=1}^{D} \tilde{\mathbf{q}}_d$
14: $\hat{\mathbf{a}} \leftarrow \text{Decoder}(\hat{\mathbf{z}})$
15: **return** reconstructed action $\hat{\mathbf{a}}$, codes $\{k_d\}_{d=1}^{D}$

---

commitment losses:

$$\mathcal{L} = \|\mathbf{x} - \mathcal{D}(\hat{\mathbf{e}})\|_2^2 + \|\text{sg}(\mathbf{e}) - \hat{\mathbf{e}}\|_2^2 + \beta\|\mathbf{e} - \text{sg}(\hat{\mathbf{e}})\|_2^2 \tag{4}$$

where $\text{sg}(\cdot)$ denotes the stop-gradient operator and $\beta$ is a hyperparameter scaling the commitment loss for multi-stage residual learning stability (Lee et al., 2022).

Due to the non-differentiability of the quantization operation $Q(\cdot)$, the straight-through estimator (STE) is employed for backpropagation by simply copying gradients from $\hat{\mathbf{e}}$ to $\mathbf{e}$ through setting $\partial\hat{\mathbf{e}}/\partial\mathbf{e} = \mathbf{I}$. This residual approach enables more precise approximation than standard VQ-VAE - with codebook size $K$ and depth $D$, it can effectively represent $K^D$ distinct quantization outputs while maintaining better computational efficiency and training stability.

**Rotation Trick.** To address the limitations of STE in preserving geometric relationships during gradient propagation, recent work(Fifty et al., 2024) proposes the rotation trick that transforms encoder outputs to codebook vectors via rotation and rescaling. For encoder output $\mathbf{e}$ and codebook vector $\mathbf{q}$, it computes:

$$\tilde{\mathbf{q}} = \frac{\|\mathbf{q}\|}{\|\mathbf{e}\|} \cdot \mathbf{R} \cdot \mathbf{e} \tag{5}$$

where $\mathbf{R}$ is the rotation matrix that aligns $\mathbf{e}$ with $\mathbf{q}$. During backpropagation, the rotation transformation preserves relative angles between gradients and vectors, enabling different points within the same quantization region to receive varying gradient updates based on their geometric relationships. This mechanism helps prevent codebook collapse and maintain diverse vector representations by encouraging appropriate exploration of the latent space.

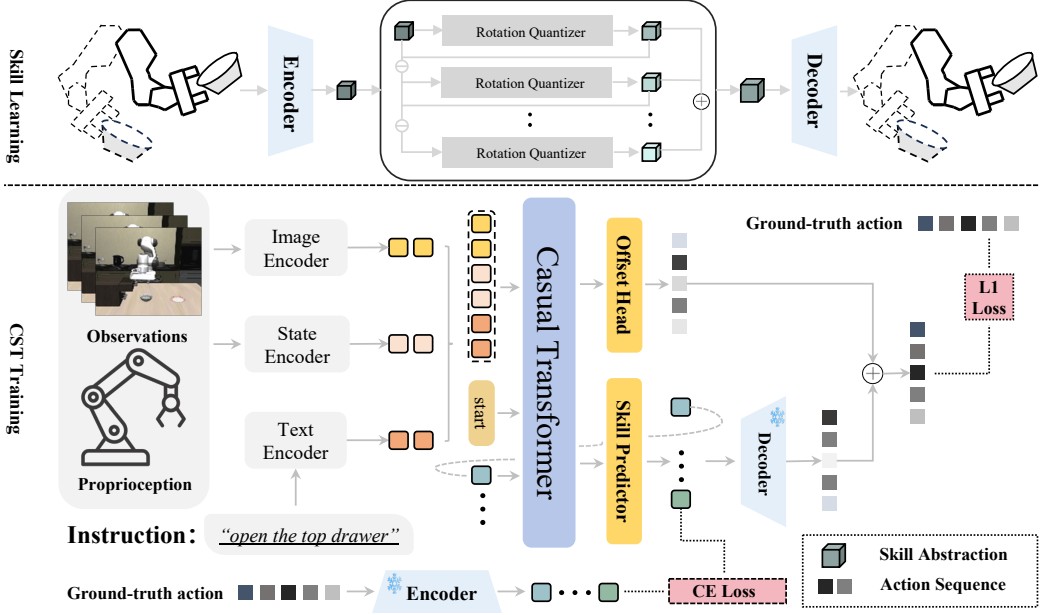

*Figure 2.* Overview of the STAR framework. Top: The RaRSQ module encodes continuous action sequences into hierarchical discrete skills through rotation-augmented residual quantization. Bottom: The CST module processes multimodal inputs (visual observations, proprioceptive states, and language instructions) to generate actions through autoregressive skill prediction and action refinement.

## 3.2. STAR

### 3.2.1. OVERVIEW

In this section, we present the Skill Training with Augmented Rotation (STAR) framework, which employs Rotation-augmented Residual Skill Quantization (RaRSQ) and a Causal Skill Transformer (CST) to enhance skill learning and composition in robotic manipulation. The framework of STAR is shown in Fig. 2. STAR adopts a two-stage training strategy: first training RaRSQ to learn skill abstractions, then fixing RaRSQ to train CST for skill composition.

### 3.2.2. ROTATION-AUGMENTED RESIDUAL SKILL QUANTIZATION

To learn expressive and reusable skill representations from continuous action sequences, we propose Rotation-augmented Residual Skill Quantization (RaRSQ). Our approach addresses two key limitations of standard VQ-VAE: codebook collapse and inefficient skill representation. By integrating rotation transformations with residual quantization, RaRSQ preserves geometric relationships between action sequences while enabling hierarchical skill encoding.

**Skill Encoding Process.** Given an action sequence $\mathbf{a}_{t:t+T}$, we first encode it into a latent vector $\mathbf{z} = \phi(\mathbf{a}_{t:t+T})$ through an encoder network $\phi$ (see Algorithm 1). RaRSQ then discretizes $\mathbf{z}$ through an iterative process as follows:

Starting with the initial residual $\mathbf{r}_0 = \mathbf{z}$, for each depth

$d = \{1, ..., D\}$, we quantize and rotate the residual:

$$k_d = \arg\min_k \|\mathbf{r}_{d-1} - \mathbf{e}_{(d,k)}\|_2^2 \qquad (6)$$

$$\tilde{\mathbf{q}}_d = \mathrm{sg}\left[\frac{\|\mathbf{e}_{(d,k_d)}\|}{\|\mathbf{r}_{d-1}\|}\mathbf{R}_d\right]\mathbf{r}_{d-1} \qquad (7)$$

$$\mathbf{r}_d = \mathbf{r}_{d-1} - \tilde{\mathbf{q}}_d \qquad (8)$$

where $\mathbf{e}_{d,k_d}$ represents the $k_d$-th vector in codebook $\mathcal{C}_d$, and $\mathrm{sg}[\cdot]$ denotes the stop-gradient operator. The rotation matrix $\mathbf{R}_d$ is computed as:

$$\mathbf{R}_d = \mathbf{I} - 2\mathbf{r}_d\mathbf{r}_d^T + 2\hat{\mathbf{q}}_d\hat{\mathbf{r}}_{d-1}^T \qquad (9)$$

where

$$\hat{\mathbf{q}}_d = \mathbf{e}_{(d,k_d)}/\|\mathbf{e}_{(d,k_d)}\|, \quad \hat{\mathbf{r}}_{d-1} = \mathbf{r}_{d-1}/\|\mathbf{r}_{d-1}\| \qquad (10)$$

$$\hat{\mathbf{r}}_d = \frac{\hat{\mathbf{r}}_{d-1} + \hat{\mathbf{q}}_d}{\|\hat{\mathbf{r}}_{d-1} + \hat{\mathbf{q}}_d\|} \qquad (11)$$

The final skill representation is obtained by summing the rotated quantized vectors:

$$\hat{\mathbf{z}} = \sum\nolimits_{d=1}^{D} \tilde{\mathbf{q}}_d \qquad (12)$$

During backpropagation, gradients flow through the rotation matrices:

$$\frac{\partial\hat{\mathbf{z}}}{\partial\mathbf{r}_{d-1}} = \frac{\|\mathbf{e}_{(d,k_d)}\|}{\|\mathbf{r}_{d-1}\|}\mathbf{R}_d \qquad (13)$$

This rotation-based gradient mechanism enables different points within the same quantization region to receive varying updates based on their geometric relationships, effectively preventing codebook collapse.

**Theoretical Benefits.** Our formulation offers three key advantages:

1. **Improved Skill Diversity:** The rotation-based gradient mechanism prevents codebook collapse by preserving geometric relationships between actions and corresponding skills. Our rotation transformation enables varied updates based on relative angular relationships, preventing skills from collapsing to a small subset of codes.
2. **Hierarchical Skill Structure:** The residual quantization naturally decomposes actions into a hierarchy, where $k_1$ captures coarse primitives while subsequent codes encode finer details, matching the inherent structure of manipulation tasks.
3. **Enhanced Representation Capacity:** The combination of residual quantization and rotation-augmented gradients enables representing $K^D$ distinct skills with only $K$ codes per level, while maintaining low quantization errors compared to naive VQ approaches.

**Training Objective.** We train RaRSQ using a combination of reconstruction and commitment losses:

$$\mathcal{L} = \mathcal{L}_{\text{recon}} + \mathcal{L}_{\text{commit}} \tag{14}$$

$$\mathcal{L}_{\text{recon}} = \|\mathbf{a}_{t:t+T} - \psi(\hat{\mathbf{z}})\|_2^2 \tag{15}$$

$$\mathcal{L}_{\text{commit}} = \beta \sum_{d=1}^{D} \| \text{sg}[\mathbf{r}_{d-1}] - \frac{\|\mathbf{e}_{d,k_d}\|}{\|\mathbf{r}_{d-1}\|} \mathbf{R}_d \mathbf{r}_{d-1} \|_2^2 \tag{16}$$

where $\psi$ is the decoder, $\text{sg}[\cdot]$ denotes stop-gradient, and $\beta$ is a weighting coefficient. The reconstruction loss $\mathcal{L}_{\text{recon}}$ ensures accurate action reconstruction, while the commitment loss $\mathcal{L}_{\text{commit}}$ encourages the residuals to stay close to their corresponding rotated and scaled skill abstractions, maintaining geometric relationships during quantization.

### 3.2.3. CAUSAL SKILL TRANSFORMER

To effectively compose learned skills for sequential manipulation tasks, we propose the Causal Skill Transformer (CST) that combines autoregressive skill prediction with adaptive refinement. Our framework explicitly models the hierarchical dependencies between skills while enabling precise action generation through refinement.

**Input Representation.** Given a sequence of observations $\mathbf{o}_{t-h:t} = \{(\mathbf{i}_k, \mathbf{p}_k)\}$, where $\mathbf{i}_k$ and $\mathbf{p}_k$ represent visual and proprioceptive inputs at timestep $k$, and a task instruction $\boldsymbol{\tau}$, we first encode the multimodal inputs using: $\mathbf{h}_k = [f_{\text{vis}}(\mathbf{i}_k); f_{\text{prop}}(\mathbf{p}_k)]$, where $f_{\text{vis}}$ and $f_{\text{prop}}$ are vision and proprioceptive encoders respectively. The transformer-based policy $\pi_\theta$ then processes these encodings along with the task embedding to generate contextual features:

$$\mathbf{g}_t = \pi_\theta([\boldsymbol{\tau}; \mathbf{h}_{t-h:t}]). \tag{17}$$

**Hierarchical Skill Prediction.** Building on our residual quantization framework, CST models skill selection as a hierarchical process where each skill depends on previous choices:

$$P(k_1, ..., k_D | \mathbf{o}_{t-h:t}, \boldsymbol{\tau}) = \prod_{d=1}^{D} P(k_d | k_{<d}, \mathbf{g}_t) \tag{18}$$

where $k_{<d}$ represents all previously predicted skill codes, and $k_d \in \{1, ..., K\}$ denotes the index selected from the $d$-th codebook. For each depth $d$, a prediction head $\zeta_d$ outputs a categorical distribution over the $K$ possible codebook indices, enabling the model to select appropriate skills at each level of abstraction. This autoregressive formulation is crucial as it captures the natural dependency structure in our residual skill space - coarse movement primitives must be selected before fine-grained adjustments.

**Action Refinement.** While the predicted codes can be directly decoded to actions using the decoder in RaRSQ, discretization of the continuous action space inevitably leads to some loss of fidelity (Shafiullah et al., 2022). Following BeT, we introduce a refinement mechanism through an offset prediction head to bridge this gap. Specifically, we add an offset head $\zeta_{\text{ref}}$ to predict continuous refinements to the discretized actions. The final action is computed as:

$$\hat{\mathbf{a}}_t = \psi(\sum_{d=1}^{D} \mathbf{R}_d \mathbf{e}_{d,k_d}) + \zeta_{\text{ref}}(\mathbf{g}_t) \tag{19}$$

where $\psi$ is the RaRSQ decoder from Section 3.2.2, $\mathbf{e}_{d,k_d}$ is the selected codebook vector at depth $d$, and $\mathbf{R}_d$ is the corresponding rotation matrix.

**Training Objective.** We optimize our framework using a combination of skill prediction and refinement losses:

$$\mathcal{L} = -\sum_{d=1}^{D} \log P(k_d^* | k_{<d}, \mathbf{g}_t) + \lambda \|\mathbf{a}_t - \hat{\mathbf{a}}_t\|^2 \tag{20}$$

where $k_d^*$ are the ground truth codes from RaRSQ encoding of the expert action $\mathbf{a}_t$, and $\lambda$ is a trade-off coefficient.

### 3.2.4. INFERENCE PROCESS

At inference time, our framework generates actions through an efficient two-stage process that balances exploration and precision. Given the current observation context $\mathbf{o}_{t-k:t}$ and task instruction $\boldsymbol{\ell}$, we perform:

**Hierarchical Skill Selection.** We first sample skill codes $(k_1, ..., k_D)$ autoregressively using nucleus sampling with temperature $\tau$ and threshold $p$:

$$k_d \sim \text{NucleusSample}(P(k_d | k_{<d}, \mathbf{g}_t), p, \tau) \tag{21}$$

**Action Generation and Execution.** The sampled skill codes are mapped to their corresponding codebook vectors and combined with predicted refinements to generate

| Method | LIBERO-Object | LIBERO-Spatial | LIBERO-Goal | LIBERO-Long | LIBERO-90 | Overall |
|---|---|---|---|---|---|---|
| Octo[†] | 85.7 ±0.9 | 78.9 ±1.0 | 84.6 ±0.9 | 51.1 ±1.3 | - | 75.1 ±0.6 |
| OpenVLA[†] | 88.4 ±0.8 | 84.7 ±0.9 | 79.2 ±1.0 | 53.7 ±1.3 | - | 76.5 ±0.6 |
| ResNet-T | 78.9 ±1.4 | 75.7 ±1.9 | 52.7 ±2.4 | 45.0 ±1.1 | 83.9 ±1.5 | 67.3 ±0.9 |
| Diffusion Policy | 62.6 ±2.8 | 69.5 ±1.8 | 54.6 ±0.5 | 51.2 ±3.0 | 75.3 ±0.7 | 62.6 ±0.6 |
| ACT | 78.8 ±1.2 | 82.0 ±0.5 | 66.1 ±1.6 | 44.0 ±0.5 | 63.4 ±5.8 | 66.8 ±1.1 |
| VQ-BeT | 90.3 ±1.5 | 88.7 ±2.0 | 61.3 ±1.0 | 59.7 ±0.2 | 84.2 ±0.3 | 76.8 ±0.5 |
| QueST | 90.0 ±1.1 | 84.5 ±0.2 | 76.7 ±0.9 | 69.1 ±1.0 | 87.4 ±0.4 | 81.5 ±0.6 |
| Ours | **98.3** ±0.2 | **95.5** ±0.6 | **95.0** ±0.7 | **88.5** ±0.3 | **90.8** ±0.2 | **93.6** ±0.1 |

Table 1. Overall Performance. The results of the baselines marked with [†] are cited from their original papers. **Bold** indicate the highest score. We report the mean and standard deviation of the normalized score with three random seeds.

actions:

$$\hat{\mathbf{a}}_{t:t+h} = \psi(\sum_{d=1}^{D} \mathbf{e}_{d,k_d}) + \zeta_{\text{offset}}(\mathbf{g}_t) \qquad (22)$$

The system executes the generated action sequence and updates observations before re-planning. This rolling horizon approach allows our framework to adapt to environment dynamics while maintaining behavioral consistency through the learned skill space.

## 4. Experiment

### 4.1. Setup and Baselines

We evaluate STAR on two comprehensive manipulation benchmarks: LIBERO (130 tasks across five suites) and MetaWorld MT50 (50 distinct manipulation tasks), plus two real-world long-horizon tasks. Success Rate (SR) is measured over 50 episodes per task with three random seeds. Detailed descriptions are in Appendix A. We compare STAR against three categories of state-of-the-art methods: (1) **Discrete LVM approaches**, (2) **End-to-end imitation learning**, (3) **Large-scale VLA models**. Detailed descriptions of these baselines can be found in Appendix A.5.1.

### 4.2. Overall Performance

As shown in Table 1, STAR significantly outperforms all baselines across different LIBERO task suites, achieving 93.6% overall success rate and surpassing the previous state-of-the-art QueST by 12.1% (81.5%). The performance improvement is particularly pronounced on LIBERO-Long tasks (88.5% vs. 69.1%, +19.4%) and complex manipulation scenarios like LIBERO-Object (98.3% vs. 90.0%, +8.3%). Compared with other baselines, STAR demonstrates consistent improvements across all task categories. For basic manipulation tasks in LIBERO-Object and LIBERO-Spatial, our method achieves 7.2%-12.6% higher success rates. The improvement margins expand significantly to 18.3%-33.9% on more challenging LIBERO-Goal and LIBERO-Long tasks.

This larger gap on complex tasks stems from the codebook

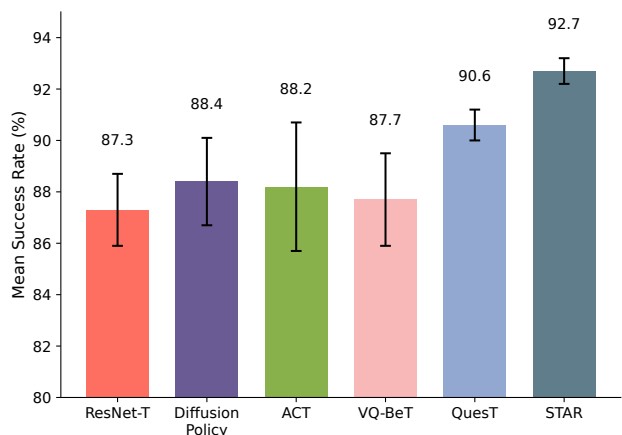

Figure 3. Performance comparison on the MetaWorld MT50 benchmark. STAR achieves consistently superior performance (92.7%) compared to baseline methods across 50 manipulation tasks.

collapse issue in discrete latent approaches like VQ-BeT and QueST. Notably, STAR even outperforms large-scale models like Octo and OpenVLA despite their access to significantly more training data.

To further validate the capability of our approach across different manipulation scenarios, we evaluate STAR on the MetaWorld MT50 benchmark. As demonstrated in Fig. 3, the strong performance extends to the MetaWorld MT50 benchmark, where STAR achieves 92.7% average success rate across all 50 tasks, outperforming existing methods with a margin of 2.1%-5.4%. The consistent improvements across both manipulation benchmarks demonstrate the effectiveness of STAR as a general framework for learning diverse robot skills.

### 4.3. Analysis of Learned Skill Diversity

To evaluate the effectiveness of STAR in preventing codebook collapse, we conduct quantitative analysis of the learned skill representations. Fig. 4 reveals several findings that demonstrate the superiority of our approach over naive VQ-VAE: First, STAR achieves complete codebook

| Method | LIBERO-Object | LIBERO-Spatial | LIBERO-Goal | LIBERO-Long | LIBERO-90 | Overall |
|---|---|---|---|---|---|---|
| Ours | **98.3** ±0.2 | **95.5** ±0.6 | **95.0** ±0.7 | **88.5** ±0.3 | **90.8** ±0.2 | **93.6** ±0.1 |
| *w/o AR* | 95.3 ±0.5 | 94.3 ±0.6 | 88.1 ±0.2 | 83.3 ±1.1 | 86.4 ±0.2 | 89.5 ±0.1 |
| *w/o Rotation* | 93.7 ±0.2 | 94.6 ±0.3 | 91.3 ±0.3 | 85.7 ±0.8 | 89.7 ±0.2 | 91.0 ±0.3 |
| *w/o Rotation and AR* | 93.3 ±0.4 | 91.9 ±0.8 | 86.9 ±0.2 | 81.5 ±1.1 | 85.5 ±2.0 | 87.8 ±0.4 |

*Table 2.* Ablation Results. "*w/o AR/Rotation*" represents removing the module of auto regressive and rotation matrix, respectively. **Best** results are marked in bold.

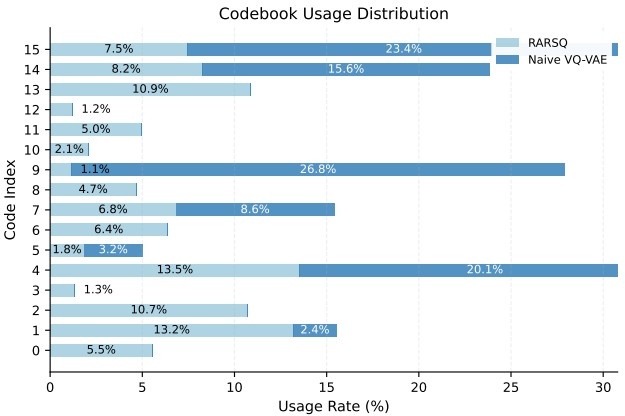

*Figure 4.* Analysis of codebook utilization patterns. Comparison between RaRSQ (light blue) and naive VQ-VAE (dark blue) approaches shows RaRSQ achieves complete codebook utilization across all 16 codes, while naive VQ-VAE exhibits severe collapse with only 7 active codes. The percentage values indicate the usage frequency of each codebook index during training.

utilization with all 16 codes being actively engaged in skill representation, while naive VQ-VAE exhibits severe collapse, utilizing only 43.8% of its codebook capacity (7 out of 16 codes). This comprehensive utilization indicates that RaRSQ successfully maintains diverse skill abstractions throughout the learning process. Second, beyond mere utilization, RaRSQ demonstrates significantly more balanced skill distribution across its codebook. The mean utilization frequency per code is 6.25%, approaching the theoretical optimal uniform distribution. In contrast, VQ-VAE shows a highly skewed distribution with 14.29% mean utilization per active code, indicating overreliance on a limited subset of representations. RaRSQ crucially maintains this healthy variation across its entire codebook rather than concentrating it in a small subset of active codes.

### 4.4. Ablation Study

To evaluate the contribution of each key component in STAR, we conduct ablation studies by removing critical modules. The results are shown in Table 2. We compare the following variants: (1) *w/o AR*: Removes the autoregressive prediction in CST, directly predicting all skill codes independently; (2) *w/o Rotation*: Removes the rotation-augmented

gradient in RaRSQ, using standard straight-through estimation; (3) *w/o Rotation and AR*: Removes both components.

The results demonstrate that both components are essential for strong performance. First, removing the autoregressive prediction (*w/o AR*) significantly impacts the ability to capture skill dependencies, leading to a substantial performance drop (89.5% vs 93.6% overall). This degradation is particularly severe on tasks requiring precise skill sequencing, such as LIBERO-Goal (-6.9%) and LIBERO-Long (-5.2%). Without autoregressive prediction, the model struggles to maintain temporal coherence between selected skills, resulting in fragmented or inconsistent behavior sequences. The rotation-augmented gradient also proves crucial for effective skill learning. Removing this component (*w/o Rotation*) leads to degraded performance (91.0% overall), with the impact most pronounced on LIBERO-Object (-4.6%). This aligns with our theoretical analysis - without rotation-augmented gradients, the model suffers from codebook collapse and fails to maintain diverse skill representations. The performance decline is especially noticeable in tasks requiring varied manipulation skills, where having a rich repertoire of distinct skills is essential.

When both components are removed (*w/o Rotation and AR*), we observe the most severe performance degradation (87.8% overall). This synergistic effect demonstrates how our two-stage approach - first learning diverse skills through rotation-augmented quantization, then capturing their causal relationships through autoregressive prediction - is crucial for effective skill learning and composition. The substantial performance gap (5.8% lower than STAR) validates our design principle of combining geometric structure preservation for skill diversity with explicit temporal dependency modeling for skill composition.

| Method | Sequential Stages | | | Overall |
|---|---|---|---|---|
| | Open | +Place | +Close | Success |
| VQ-BET | 4/10 | 3/10 | 1/10 | 1/10 |
| QueST | 3/10 | 1/10 | 0/10 | 0/10 |
| Ours | 6/10 | 4/10 | 3/10 | 3/10 |

*Table 3.* Performance on sequential manipulation task (drawer operation). Results show successful trials out of 10 attempts for each stage and overall task completion.

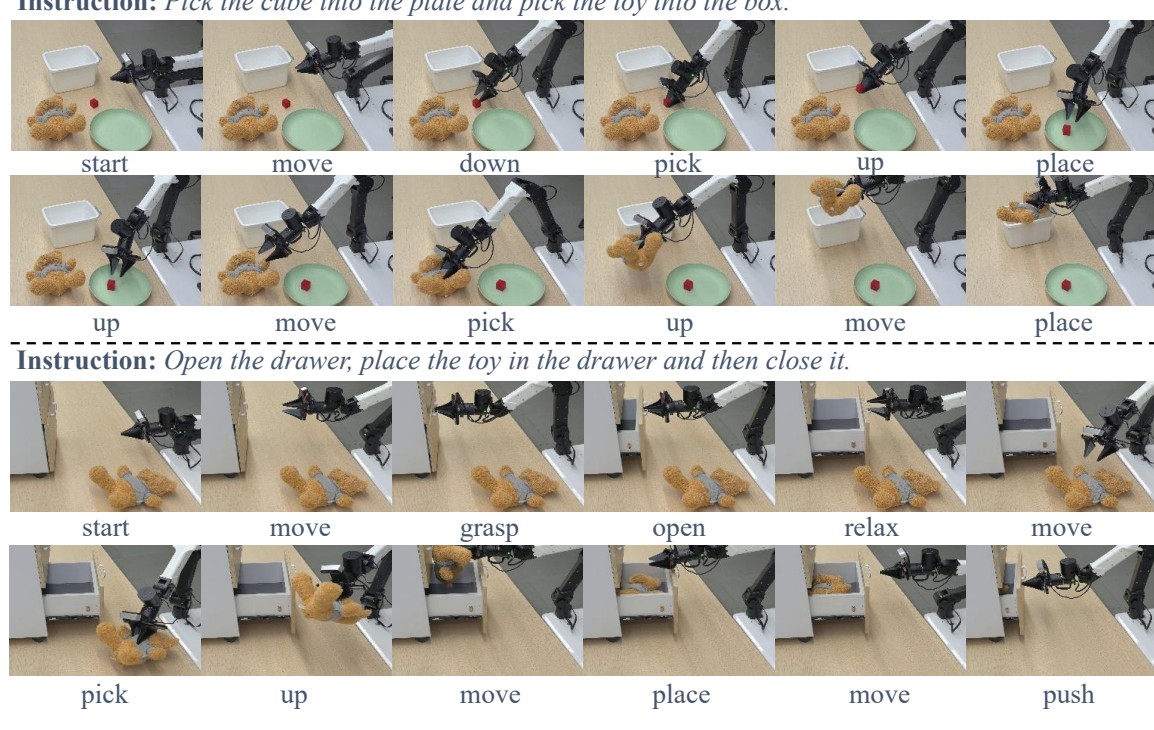

**Instruction:** *Open the drawer, place the toy in the drawer and then close it.*

*Figure 5.* Visualization of two real-world manipulation tasks with key execution stages. Each frame shows a critical step in the manipulation sequence.

| Method | Task Completion | | Overall |
| --- | --- | --- | --- |
| | Cube→Plate | Toy→Box | Success |
| VQ-BET | 5/10 | 3/10 | 3/10 |
| QueST | 6/10 | 4/10 | 4/10 |
| Ours | 8/10 | 6/10 | 6/10 |

*Table 4.* Performance on sequential manipulation task (sequential object placement). Results show successful trials out of 10 attempts for each subtask and overall completion.

### 4.5. STAR on Real-World Robots

To validate the effectiveness of STAR beyond simulation, we evaluate our approach on two challenging real-world manipulation tasks: a sequential object placement task ("Pick the cube into the plate and pick the toy into the box") and a structured drawer manipulation sequence ("Open the drawer, place the toy in the drawer and then close it"). These tasks mirror the complexity of LIBERO-Long tasks while introducing real-world challenges like lighting variations and physical dynamics.

For the drawer manipulation task, as shown in Table 3, the results highlight a key advantage of the hierarchical skill decomposition. While all methods can initiate basic actions like drawer opening (60% success rate for STAR), performance degrades through complex sequences. STAR maintains higher success rates across stages, achieving 30%

complete task success compared to 10% for VQ-BeT and 0% for QueST. The performance degradation from opening (60%) to complete execution (30%) reveals the compounding difficulty of maintaining precise control through extended sequences, though the degradation of STAR is notably less severe than the baselines.

As show in Table 4, the sequential object placement results further demonstrate the effectiveness of STAR in handling varied manipulation skills. STAR achieves 60% success rate for complete task execution, significantly outperforming VQ-BeT and QueST. The performance difference between initial cube placement and the more challenging toy placement aligns with task complexity, as the second placement requires more precise control given the confined space of the box. These results, visualized in Fig. 5, demonstrate that the improvements of STAR in skill learning and composition are effective in real-world scenarios.

## 5. Conclusion

In this paper, we presented STAR, a framework for learning and composing diverse robot skills through rotation-augmented vector quantization. Through RaRSQ and CST, our approach effectively prevents codebook collapse while enabling precise skill composition. Extensive experiments demonstrate the strong performance of STAR across manip-

ulation benchmarks, significantly outperforming state-of-the-art methods in both simulation and real-world settings.

**Limitations.** While STAR demonstrates strong performance across benchmarks, our approach requires predefined codebook sizes and quantization depths, which must be manually tuned for different task domains. Further, as an imitation learning approach, STAR depends on the quality of available expert demonstrations, which may limit its applicability when such data is scarce.

## Acknowledgements

We would like to thank all co-authors for their efforts and the reviewers for their constructive comments. This work is supported by National Natural Science Foundation of China (Grant No. 62306090), National Natural Science Foundation of China (Grant No. 62406092), National Natural Science Foundation of China (Grant No. U24B20175), Natural Science Foundation of Guangdong Province of China (Grant No. 2024A1515010147), Guangdong Basic and Applied Basic Research Foundation (Grant No. 2025A1515010169), Shenzhen Science and Technology Program (Grant No. KQTD20240729102207002), Shenzhen Science and Technology Program (Grant No. KJZD20240903100017022), and Research on Efficient Exploration and Self-Evolution of APP Agents & Embodied Intelligent Cerebellum Control Model and Collaborative Feedback Training Project (Grant No. TC20240403047).

## Impact Statement

This paper presents work whose goal is to advance the field of Machine Learning. There are many potential societal consequences of our work, none which we feel must be specifically highlighted here.

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

# A. Experimental and Dataset

Put the yellow and white mug in the microwave and close it demo

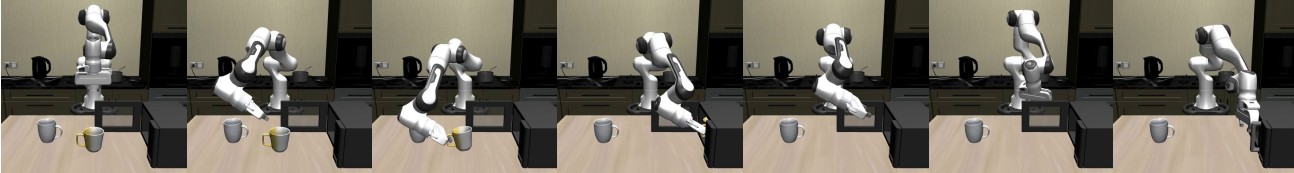

turn on the stove and put the moka pot on it demo

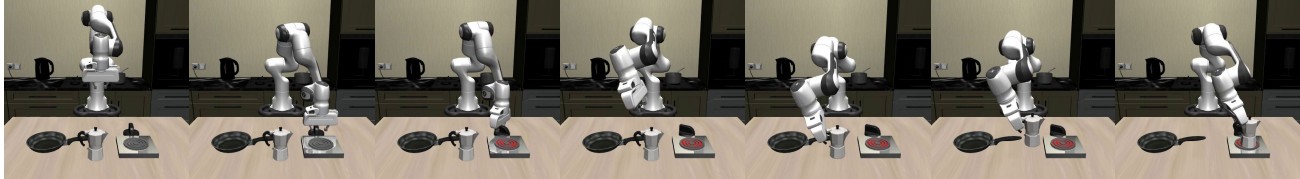

put both the alphabet soup and the cream cheese box in the basket demoput the moka pot on it demo

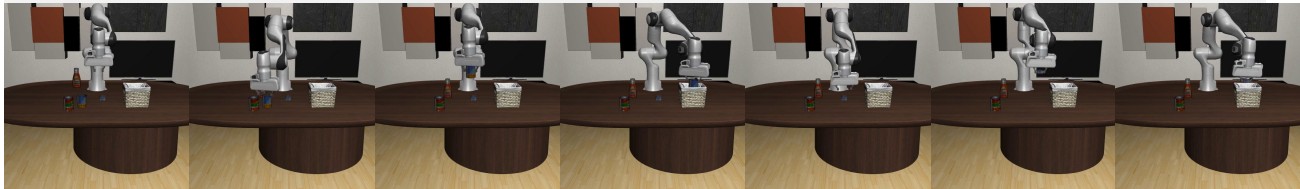

*Figure 6.* The visualization of simulated tasks, including

## A.1. Benchmark Environments

We evaluate STAR across multiple comprehensive benchmarks spanning both simulated and real-world environments to assess its effectiveness in diverse manipulation scenarios.

### A.1.1. LIBERO BENCHMARK

LIBERO is a comprehensive benchmark for language-conditioned manipulation tasks that evaluates different aspects of robotic manipulation capabilities through five distinct task suites:

- **LIBERO-Spatial** contains 10 tasks focusing on identical objects in different spatial layouts, testing the understanding of spatial relationships.

- **LIBERO-Object** comprises 10 tasks with consistent layouts but varying objects, evaluating the ability to generalize across different object types.

- **LIBERO-Goal** features 10 tasks sharing the same object categories and spatial layouts but with different goals, assessing the capability to learn diverse task-oriented behaviors.

- **LIBERO-Long** presents 10 challenging long-horizon tasks involving diverse object categories and layouts, testing temporal reasoning and sequential manipulation skills.

- **LIBERO-90** encompasses 90 tasks with extremely diverse object categories, layouts, and task goals, providing a comprehensive evaluation platform.

For our experiments, we utilize 50 expert demonstrations per task from the author-provided dataset. These demonstrations capture diverse manipulation strategies and initial conditions, enabling robust policy learning.

### A.1.2. METAWORLD MT50

MetaWorld MT50 is a challenging multi-task benchmark consisting of 50 distinct manipulation tasks performed by a Sawyer robotic arm in simulation. The tasks range from basic object manipulation (e.g., pushing, pulling) to complex tool usage (e.g., using a hammer, operating a door lock). Each task requires precise control to achieve specific goals, such as moving objects to target locations or manipulating articulated objects to desired configurations. The MT50 setting specifically evaluates the ability to simultaneously learn and master all 50 tasks, making it a rigorous test for skill learning and composition.

For training, we collect 100 demonstrations per task using the scripted policies provided in the official MetaWorld codebase. These policies ensure consistent and optimal task execution, providing a reliable basis for learning manipulation skills.

## A.2. Real-World Setup

To validate the effectiveness of our approach in real-world scenarios, we conducted experiments using the Cobot Agilex ALOHA robot, a dual-arm manipulator platform. For our experiments, we utilized a single arm to perform two challenging sequential manipulation tasks designed to test the ability to handle complex, multi-stage operations.For each task, we collected 45 demonstrations through human teleoperation.

### A.2.1. SEQUENTIAL OBJECT PLACEMENT TASK

The first task required the robot to "Pick the cube into the plate and pick the toy into the box." This task was designed to evaluate the ability to perform sequential pick-and-place operations involving multiple objects and target locations. Specifically, the robot needed to:

- Successfully grasp and place a cube onto a plate.

- Transition to a second object (a toy) and place it into a box.

### A.2.2. DRAWER MANIPULATION TASK

The second task involved a more complex sequence: "Open the drawer, place the toy in the drawer, and then close it." This task was designed to assess the capability to handle confined spaces and sequential manipulation steps. The robot needed to:

- Open a drawer.

- Place a toy inside the drawer.

- Close a drawer.

## A.3. Evaluation Protocol

To ensure a rigorous evaluation of our approach, we adopted a comprehensive evaluation protocol across both simulated and real-world tasks.

### A.3.1. SIMULATED BENCHMARKS

For the LIBERO and MetaWorld MT50 benchmarks, we evaluated performance using the Success Rate (SR) metric, calculated over 50 episodes per task with three random seeds. This protocol allowed us to assess the robustness and consistency of our approach across different task configurations.

### A.3.2. REAL-WORLD TASKS

For the real-world tasks, we conducted 10 trials per task and reported both the overall success rate and the stage-wise completion rate. Specifically:

- For the sequential object placement task, we tracked the successful completion of both the cube-to-plate and toy-to-box placements.

- For the drawer manipulation task, we measured success rates across three sequential stages: drawer opening, toy placement, and drawer closing.

A trial was considered successful only if all stages of the task were completed in the correct sequence. This evaluation protocol allowed us to identify potential bottlenecks in the manipulation sequence and assess the overall robustness of the learned policies.

### A.4. Implementation Details

we provide comprehensive implementation details of our framework, including the architecture configurations and training settings for both RaRSQ and CST modules, as well as the optimization process.

For the **Rotation-augmented Residual Skill Quantization (RaRSQ)** module in our proposed STAR framework, we adopt a single-layer MLP as the encoder with a hidden dimension of 128. The decoder is implemented as a transformer with 4 attention heads, 4 decoder layers, and a hidden dimension of 128. We set the codebook size $K = 16$ and the quantization depth $D = 2$, with each skill abstraction spanning 8 timesteps. During training, the encoder and decoder are jointly optimized, while the codebook vectors are updated through the rotation-augmented gradient mechanism.

For the **Causal Skill Transformer (CST)**, we utilize a ResNet-18 model trained from scratch as the visual encoder and a pre-trained CLIP-base model as the language encoder. The proprioception encoder is implemented as a single-layer MLP with a hidden dimension of 128. The transformer decoder consists of 6 layers, 6 attention heads, and an embedding dimension of 384. We set the start token dimension to 16, the beam size to 5, and the temperature to 1.0 for sampling. The observation window is fixed at 10 timesteps. During training, the visual and language encoders are frozen, and only the weights of the transformer decoder and the offset prediction head are updated.

We train the entire framework using the AdamW optimizer with a cosine decay learning schedule. For RaRSQ module, we use a batch size of 1024, learning rate of $5.5e$-5, and train for 100 epochs. For CST module, we use a batch size of 512, learning rate of $8e$-4, and train for 500 epochs. Both modules use a warmup step of 10 epochs and weight decay of $1e$-6. The loss weights for the first codebook prediction, second codebook prediction, and offset head prediction are set to 2.0, 1.0, and 20.0, respectively. The models are implemented in PyTorch and trained on a server with 8 Nvidia RTX L40S 48GB GPUs, with all models easily fitting on a single GPU.

### A.5. Baseline Implementation

#### A.5.1. BASELINE DESCRIPTION

We systematically evaluate STAR against state-of-the-art methods in three major categories:

**Discrete LVM approaches:** (1) **VQ-BeT** (Lee et al., 2024) combines VQ-VAE for discrete latent space learning with a transformer for latent code prediction; (2) **QueST** (Mete et al., 2024) leverages Finite-State Quantization for discrete latent space construction and employs a causal transformer for action sequence modeling.

**End-to-end imitation learning:** (1) **ResNet-T** (Liu et al., 2024) uses ResNet-18 with FiLM for observation and task instruction encoding, followed by a transformer and GMM output layer for action prediction; (2) **Diffusion Policy**(Chi et al., 2023) implements a UNet-based architecture that maps Gaussian noise to action trajectories through a learned denoising diffusion process; (3) **ACT** (Zhao et al., 2023) proposes a transformer-based CVAE that generates temporally extended action sequences by decomposing behaviors into action chunks.

**Large-scale VLA models:** (1) **Octo** (Team et al., 2024) features a large-scale transformer policy trained on 800K robotic demonstrations with diffusion-based action generation; (2) **OpenVLA** (Kim et al., 2024) presents a 7B-parameter vision-language-action model that integrates Llama 2 with DINOv2 and SigLIP visual features.

#### A.5.2. BASELINE IMPLEMENTATION

Following prior works, we compare our method with several representative approaches in robotic manipulation. To ensure fair comparison, these baselines are implemented with consistent input modalities and encoders.

**ResNet-T.** The architecture consists of a transformer with 6 layers and hidden dimension of 256. The observation encoder incorporates ResNet-18 features combined with spatial attention mechanisms, maintaining a temporal context window of 10 timesteps for sequential prediction.

**ACT.** The architecture incorporates 8 cross-attention and 4 self-attention layers, processing visual inputs through a ResNet-18 backbone and language inputs via a CLIP text encoder. The continuous action space is discretized into 256 bins per dimension, with model optimization performed using AdamW and cosine learning rate decay.

**Diffusion Policy.** The backbone follows a U-Net design with channel dimensions [256, 512, 1024]. For the LIBERO benchmark, the prediction and execution horizons are set to $T = 32$ and $T_a = 16$ respectively, while the MetaWorld setup uses $T = 16$ and $T_a = 8$. The policy maintains single-step observation history during execution.

**VQ-BeT.** The framework employs a single-layer MLP encoder (dimension 128) and a residual vector quantization module with approximately 1024 codes. The sequence processing utilizes an observation window of 10 timesteps and action window size $T = 5$, as longer sequences ($T = 32$) lead to information loss through excessive compression.

**QueST.** The model utilizes a transformer decoder with 6 layers and 6 attention heads (embedding dimension 384). The action generation process incorporates a vocabulary size of 1000 and equivalent start token dimension. At inference time, beam search is applied with width 5 and temperature 1.0, operating on action blocks of 8 timesteps.

For all baselines, multi-modal observations are processed through concatenation and dimension-specific projections, with hyperparameters following their respective original implementations.

## B. Extensive Ablation Studies

| Method | LIBERO-Object | LIBERO-Spatial | LIBERO-Goal | LIBERO-Long | LIBERO-90 | Overall |
|---|---|---|---|---|---|---|
| Ours | **98.3** ±0.2 | **95.5** ±0.6 | **95.0** ±0.7 | **88.5** ±0.3 | **90.8** ±0.2 | **93.6** ±0.1 |
| *w/o Action Refinement* | 87.7 ±1.7 | 76.8 ±2.8 | 54.4 ±0.5 | 37.6 ±1.6 | 38.4 ±0.3 | 59.0 ±0.8 |

*Table 5.* Ablation Results. "*w/o Action Refinement*" represents removing the module of action refinement. **Best** results are marked in bold.

### B.1. Ablation Study on Action Refinement

To investigate the importance of action refinement in our framework, we conduct ablation experiments by removing the refinement module while keeping all other components unchanged. As shown in Table 5, removing action refinement leads to substantial performance degradation across all task suites.

The impact is particularly pronounced on more complex tasks that require precise manipulation. For LIBERO-Long tasks, removing action refinement causes a dramatic drop in performance from 88.5% to 37.6% (-50.9%). Similarly, for LIBERO-Goal tasks, the success rate decreases from 95.0% to 54.4% (-40.6%). This significant performance gap highlights that discrete skill codes alone, while effective for capturing high-level behavior patterns, are insufficient for achieving the precision required in complex manipulation tasks.

Even for relatively simpler tasks in LIBERO-Object and LIBERO-Spatial, we observe notable performance decreases of 10.6% and 18.7% respectively. This suggests that action refinement plays a crucial role in bridging the gap between discrete skill abstractions and continuous control requirements, even for basic manipulation tasks.

The impact becomes more severe as task complexity increases, as evidenced by the larger performance drops in LIBERO-90 (-52.4%) and LIBERO-Long (-50.9%). This trend can be attributed to two factors:

1. Complex tasks often require more precise adjustments to handle varying object positions and environmental conditions, which cannot be fully captured by discrete skill codes alone.

2. Longer manipulation sequences accumulate small precision errors from discrete quantization, making the refinement mechanism increasingly important for maintaining task success over extended horizons.

Overall, the ablation results demonstrate that action refinement is an essential component of our framework, contributing to a 34.6% improvement in average performance across all tasks. This validates our design choice of combining discrete skill abstractions with continuous refinement to enable both structured behavior representation and precise control execution.

## C. Extensive Analysis

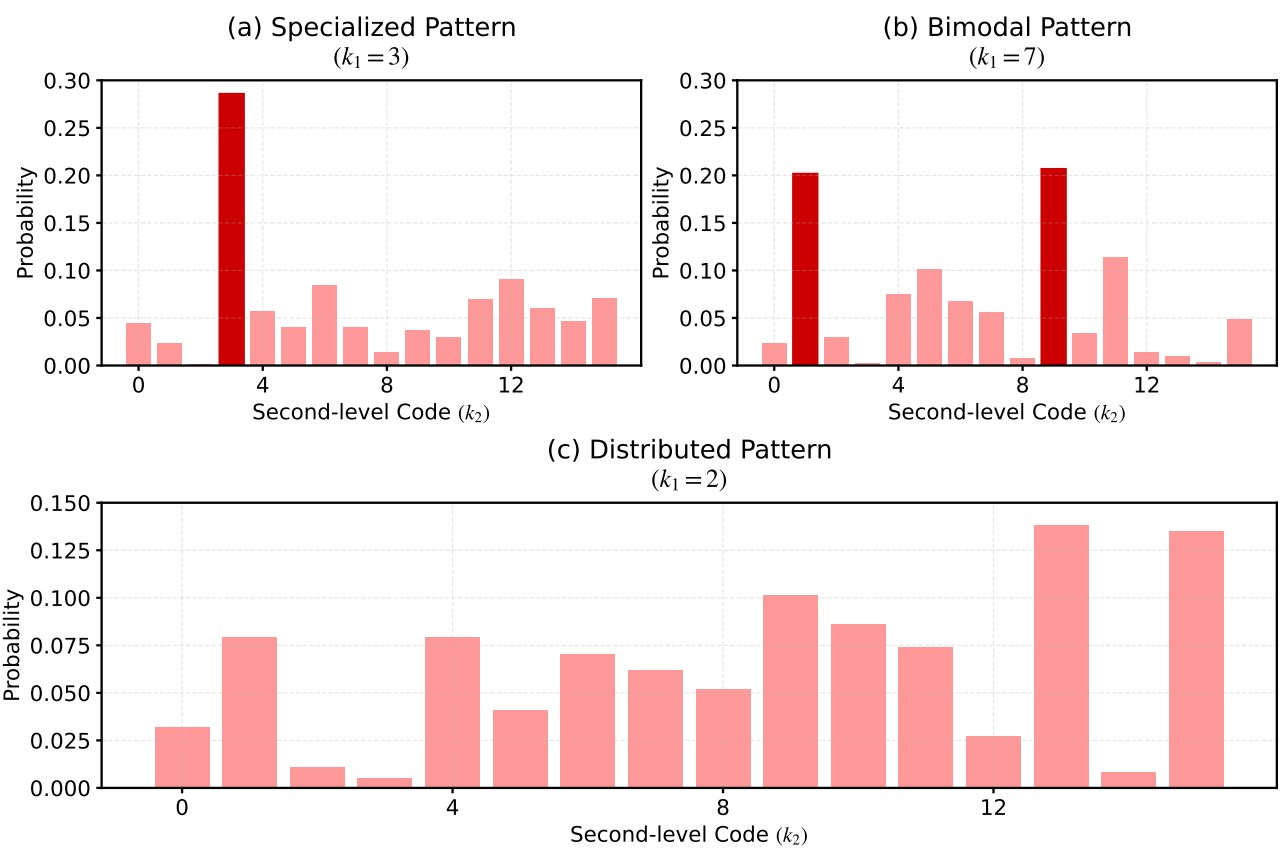

*Figure 7.* Visualization of representative skill dependency patterns. (a) Specialized pattern showing strong preference for specific second-level codes ($k_1$=3). (b) Bimodal pattern indicating branching skill relationships ($k_1$=7). (c) Distributed pattern with more uniform dependencies ($k_1$=2).

### C.1. Evaluation of Skill Composition

We analyze the conditional probability distribution $P(k_2|k_1)$ between first and second codebook indices to understand the hierarchical dependencies in our skill abstraction approach. Fig. 7 visualizes three representative patterns that demonstrate how our framework decomposes complex behaviors.

As shown in Fig. 7(a), when $k_1 = 3$, we observe a highly concentrated distribution with a prominent peak at $k_2 = 3$ ($P(k_2 = 3|k_1 = 3) = 0.286$), indicating this first-level skill code consistently pairs with specific second-level codes. Fig. 7(b) shows a bi-modal distribution for $k_1 = 7$, suggesting this primitive skill branches into two distinct paths. In contrast, Fig. 7(c) illustrates a more uniform distribution for $k_1 = 2$, indicating more flexible skill combinations.

These diverse dependency patterns validate our hierarchical skill decomposition design, demonstrating that RaRSQ effectively captures different aspects of manipulation behaviors at multiple abstraction levels. This structured representation enables CST to generate coherent action sequences for accurate control.

### C.2. Codebook Relationship

The relationship between consecutive codebook layers in our RaRSQ approach reveals important insights into how robot skills are hierarchically represented and composed. Fig. 8 visualizes the conditional probability distribution $P(k_2|k_1)$ between indices from the first and second codebooks, demonstrating several key patterns that validate our architectural choices.

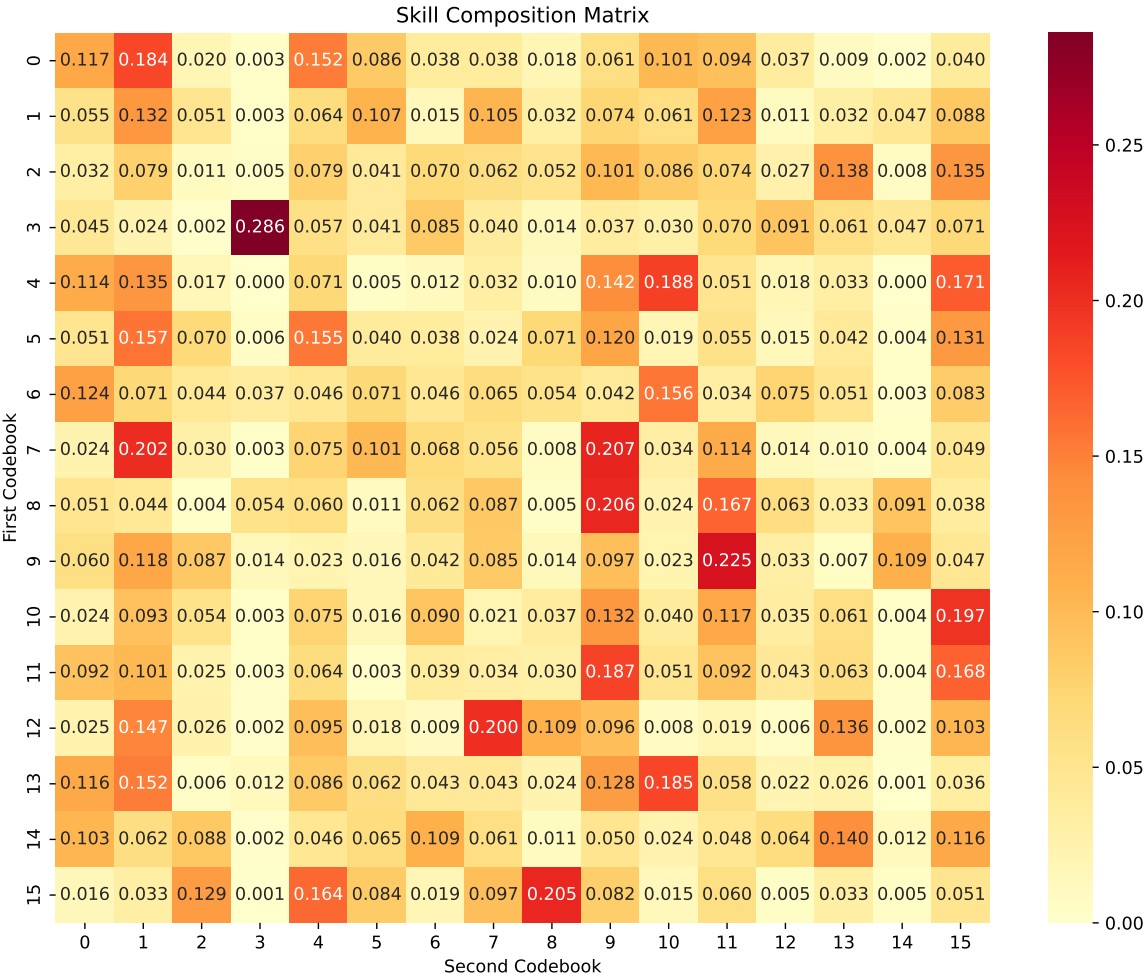

*Figure 8.* Visualization of conditional probability distribution $P(k_2|k_1)$ between first-level ($k_1$) and second-level ($k_2$) codebook indices.

The heatmap exhibits distinct non-uniform distributions across different first-level codes, indicating strong statistical dependencies between the two codebook levels. For instance, when $k_1 = 3$, we observe a particularly high probability of $P(k_2 = 3|k_1 = 3) = 0.286$, suggesting this second-level code frequently serves as a refinement for the third first-level primitive skill.

We observe several interesting patterns in the codebook relationships:

1. **Specialized Connections:** Some first-level codes show strong preferences for specific second-level codes. For example, $k_1 = 7$ exhibits high probabilities with $k_2 = 1$ and $k_2 = 9$ ($P(k_2 = 1|k_1 = 7) = 0.202$ and $P(k_2 = 9|k_1 = 7) = 0.207$), indicating these pairs may capture complementary aspects of certain manipulation skills.

2. **Distributed Patterns:** Other first-level codes (e.g., $k_1 = 2$) demonstrate more uniform distributions across second-level codes, suggesting these represent more general behaviors requiring diverse refinements.

3. **Balanced Utilization:** The moderate maximum conditional probability (0.286) and the presence of both strong (dark red) and weak (light yellow) connections indicate the model learns meaningful skill hierarchies while avoiding over-specialization.

These patterns validate several key design choices in our framework:

- The clear statistical dependencies between codebook levels support our use of autoregressive prediction in CST

- The balanced utilization patterns demonstrate the effectiveness of RaRSQ in preventing codebook collapse

- The hierarchical structure revealed by the conditional probabilities confirms effective decomposition of complex behaviors into primitive skills and their refinements

This analysis provides quantitative evidence for both the effectiveness of our hierarchical skill encoding and the importance of modeling dependencies between codebook levels for robust skill composition.

## C.3. Skill Quantization Loss

To further validate the effectiveness of our rotation-augmented residual skill quantization (RaRSQ) approach, we analyze the quantization loss across different LIBERO task suites. The quantization loss measures the L1 distance between the encoder outputs and their quantized representations, serving as a direct indicator of how well the codebook captures the underlying skill space.

Fig. 9 shows the evolution of quantization loss during training for both standard VQ-VAE and our RaRSQ approach. Several key observations emerge from this comparison:

- **Initial Convergence:** Both approaches start with similar loss values (around 0.001) across all task suites, indicating comparable initialization conditions. However, RaRSQ demonstrates consistently lower initial losses (e.g., 0.00116 vs 0.00121 for LIBERO-Object at step 100), suggesting better initialization of the codebook structure.

- **Training Dynamics:** Standard VQ-VAE exhibits a concerning pattern where the quantization loss increases substantially during training before partial recovery. For instance, in LIBERO-Spatial, the loss peaks at 0.0105 around step 5300 before settling at 0.0094. This pattern is indicative of codebook collapse, where the model struggles to maintain diverse skill representations.

- **Stability:** In contrast, RaRSQ shows remarkably stable training dynamics. While it experiences minor initial increases in loss (e.g., peaking at 0.004 for LIBERO-Spatial around step 600), these increases are both smaller in magnitude and shorter in duration compared to standard VQ-VAE.

- **Final Performance:** RaRSQ achieves significantly lower final quantization losses across all task suites. The improvements are particularly pronounced for complex tasks - LIBERO-Long shows a final loss of 0.00025 with RaRSQ compared to 0.0076 with standard VQ-VAE, representing a 30× reduction in quantization error.

The loss patterns strongly correlate with our main experimental findings. The lower and more stable quantization losses of RaRSQ directly translate to improved task performance, particularly for complex manipulation sequences where precise skill representation is crucial. The brief initial increase in loss likely represents an exploration phase where the model discovers and refines its skill abstractions, while the subsequent rapid convergence to low loss values indicates successful preservation of geometric relationships through our rotation-augmented gradient mechanism.

Notably, the reduction in quantization error is most significant for LIBERO-Long (97% reduction) and LIBERO-90 (92% reduction), precisely the task suites where our method shows the largest performance improvements. This suggests that the ability of RaRSQ to maintain low quantization error is particularly beneficial for complex, long-horizon tasks that require precise and diverse skill representations.

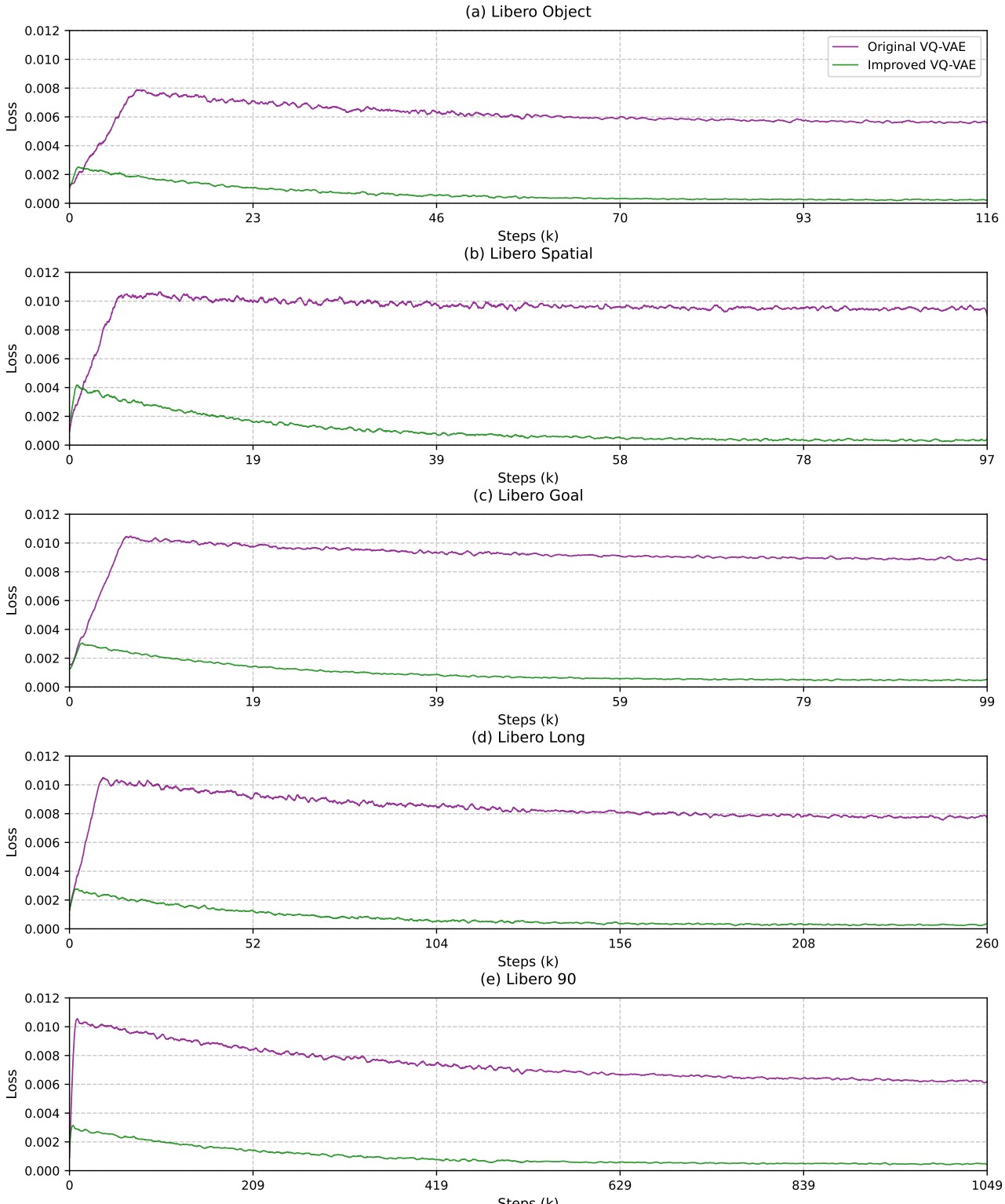

*Figure 9.* skill quantization loss

