# OpenReview forum: "STAR: Learning Diverse Robot Skill Abstractions through Rotation-Augmented Vector Quantization"
_ICML.cc/2025/Conference — ICML 2025 spotlightposter_

### Official Review · Reviewer_6m5A · 2025-03-05

**Overall Recommendation:** 3

**Summary:**

This paper proposes STAR, a novel framework for learning diverse robot manipulation skills through skill quantization and causal modeling. STAR consists of two key components: RaRSQ, which enhances residual skill quantization with a rotation-based gradient mechanism to mitigate codebook collapse, and CST, a transformer-based model explicitly capturing temporal dependencies between discrete skills. The authors demonstrate that STAR significantly improves performance in multi-task imitation learning and complex, long-horizon manipulation tasks on standard benchmarks, achieving state-of-the-art results and highlighting its effectiveness in accurately composing complex action sequences.

**Claims And Evidence:**

Yes

**Essential References Not Discussed:**

No

**Experimental Designs Or Analyses:**

I find the experimental designs reasonable and sufficiently aligned with the paper’s objectives.

**Methods And Evaluation Criteria:**

Yes

**Other Comments Or Suggestions:**

To better highlight the strengths of your proposed approach, it would be beneficial to compare the performance explicitly against QueST with an additional offset prediction mechanism. This would clarify whether the observed performance improvements truly stem from the proposed combination or primarily from the offset predictor component.

**Other Strengths And Weaknesses:**

Strengths:

The paper effectively integrates existing techniques (Residual Quantization, Rotation Trick, and causal transformer architectures) into a  practically effective framework.

The experimental evaluation is extensive, utilizing multiple benchmarks (LIBERO, MetaWorld).

Weaknesses:

While the integration of existing techniques is practically effective, the proposed methodologies heavily rely on adaptations and combinations of previously established ideas, which somewhat limits the theoretical novelty.

Specifically, the proposed approach significantly depends on well-established techniques (residual quantization and the rotational trick), and autoregressive skill prediction has already been extensively explored in QueST. My primary concern is that the notable performance improvement observed in this paper may arise more from offset prediction, as indicated in Table 5, rather than the novel combination itself.

**Questions For Authors:**

Have the authors considered comparing their method explicitly against QueST equipped with an offset prediction mechanism to isolate the contributions of the proposed combination more clearly?

It is currently unclear which component of proposed method is primarily responsible for the observed performance gains. Could you further clarify to highlight the contribution of each component more clearly?

**Relation To Broader Scientific Literature:**

This paper builds upon recent advances in residual quantization methods and latent variable models (e.g., VQ-VAE, Rotation Trick) for robot skill learning, addressing common limitations such as codebook collapse and insufficient temporal skill modeling by introducing a novel combination of rotation-based quantization (RaRSQ) and autoregressive causal skill modeling (CST).

**Theoretical Claims:**

No theoretical section is presented in the paper.

---

> ### Author Rebuttal · Authors · 2025-04-01
>
> Thank you for your thoughtful review. We appreciate your recognition of our extensive experiments and the practical effectiveness of our approach. We address your questions below:
> ## Q1：Have the authors considered comparing their method against QueST equipped with an offset prediction to isolate the contributions of the proposed combination more clearly?
> We appreciate and follow your good advice by adding additional experiments that compare STAR against QueST across various configurations:
> |Method|Finetune decoder|Offset|LIBERO-Object|LIBERO-Spatial|LIBERO-Goal|LIBERO-Long|Avg|
> |-|-|-|-|-|-|-|-|
> |QueST (Original)|✅|❌|90.0|84.5|76.7|69.1|80.1|
> |QueST(freeze decoder)|❌|❌|78.1|63|56|24|55.3|
> |QueST(fine-tune decoder + offset)|✅|✅|85.8|73.9|65.9|64.7|72.6|
> |Ours|❌|✅|**98.3**|**95.5**|**95.0**|**88.5**|**94.3**|
>
> The results demonstrate that QueST with offset prediction (72.6%) significantly underperforms compared to our STAR method (94.3%)
>
> We introduce offset head because our decoder is frozen during stage-1 training. This design means predicted actions can only reconstruct stage-0 actions, which undergo lossy compression through quantization, making fine-grained operations difficult. Therefore, we use an additional offset head to compensate for this gap.
>
> In contrast, quest has already fine-tuned the decoder to learn fine-grained operations, which serves a similar purpose to our approach but through a different mechanism. Further adding an offset head does not yield significant gains and may instead conflict with the decoder's output.
>
> To further verify the importance of learning fine-grained operations, we conducted an ablation study with QueST using a frozen decoder in stage-1. Performance dropped from 80.1% to 55.3%, consistent with our findings in Table 5 where removing the offset head significantly reduces our method's performance.
>
> In summary, both approaches require mechanisms for learning fine-grained operations, but the superior performance of STAR comes from our novel contributions (RaRSQ and CST) rather than the offset head alone. The ablation studies in our original Table. 2 further support this conclusion by isolating the contributions of each proposed component.
> ## Q2：It is currently unclear which component is primarily responsible for the performance gains. Could you further clarify to highlight the contribution of each component more clearly?
> Our framework consists of two components: (1)rotation-augmented residual skill quantization (RaRSQ), which addresses codebook collapse in robotic skill learning, and (2)causal skill transformer (CST) , which captures causal relationship between skills through autoregressive prediction. We would like to further clarify the contribution of each component:
> ### **RaRSQ: Enhanced Skill Diversity and Representation**
> RaRSQ directly addresses the fundamental limitation of naive residual VQ-VAE - codebook collapse. As shown in Fig. 4, residual VQ-VAE utilizes only 43.8% codes, severely limiting the robot's ability to express diverse actions. In contrast, RaRSQ achieves 100% codebook utilization with balanced distribution.
>
> This enhanced skill diversity translates to performance gains through precise skill decomposition. Complex manipulation tasks inherently require fine-grained action representation. With full codebook utilization, RaRSQ can distinguish between subtly different manipulation skills (e.g., picking different objects, precise positioning) that would otherwise be mapped to the same code in collapsed codebooks. This enhances the model's capability to recognize and execute task-specific manipulation patterns.
> ### **CST's Contribution to Skill Composition**
> CST's autoregressive design comes from analyzing the learned hierarchical dependencies between skills. The conditional probability analysis in Fig. 8 reveals causal relationships between first and second-level skills -  some first-level codes show strong preferences for specific second-level codes. CST models the causal dependencies between skills, which is crucial for generating coherent action sequences. As show in Table. 2, when removing CST, we observe a significant performance drop, especially for LIBERO-Goal (-6.9%) and LIBERO-Long (-5.2%), indicating that modeling these dependencies is particularly important for complex manipulation tasks.
> ## W1：While the integration of existing techniques is practically effective, the proposed method rely on adaptations and combinations of established ideas, which somewhat limits the theoretical novelty.
> Our key contributions lie in designing a robot-specific residual VQ-VAE that effectively decomposes complex robot behaviors into discrete skills, addressing fundamental challenges in robot skill learning. RaRSQ directly addresses codebook collapse in robotic residual skill quantization, while CST explicitly models the causal dependencies between different skill abstraction levels, revealing structured relationships between coarse and fine-grained behaviors.

---

> > ### Comment · Reviewer_6m5A · 2025-04-05
> >
> > Thank you for your detailed rebuttal.
> >
> > I especially appreciate the additional experimental comparisons provided, clearly distinguishing the contributions of STAR from the other baseline methods. Considering the additional experiments and context provided, I now have a more positive view of this work.

---

> > > ### Author Response · Authors · 2025-04-05
> > >
> > > Thank you once again for your positive view of our work. We truly appreciate your valuable feedback.

---

### Official Review · Reviewer_Lv7s · 2025-03-16

**Overall Recommendation:** 4

**Summary:**

The paper investigates robot skill abstraction for manipulation tasks and introduces STAR—a framework for learning discrete robot skill representations. STAR comprises two main components: Rotation-Augmented Residual Skill Quantization (RaRSQ), which mitigates codebook collapse in VQ-VAE-based methods using rotation-based gradients, and a Causal Skill Transformer (CST) that explicitly models dependencies between hierarchical skill representations via an autoregressive mechanism. Experiments on the LIBERO and MetaWorld MT50 benchmarks, as well as real-world tasks, demonstrate that STAR outperforms several baselines in terms of success rate.

**Claims And Evidence:**

The paper makes two key claims. First, the proposed RaRSQ method helps mitigate codebook collapse in VQ-VAE-based skill abstraction (quantization) by leveraging rotation-based residual skill abstraction. Second, the proposed causal skill transformer explicitly models dependencies between skill representations through an autoregressive mechanism, making it effective for complex, long-horizon manipulation tasks. To validate these claims, the authors conducted experiments on the LIBERO and MetaWorld MT50 benchmarks, as well as real-world experiments. Results of experiments demonstrate superior performance of the method and thus support the claim.

**Essential References Not Discussed:**

A broader discussion on recent progress in robot learning—especially work on vision-language-action (VLA) transformers like π0 [1]—would provide valuable context regarding performance, efficiency, and scalability.

[1] Kevin Black, et al.,”π0: A Vision-Language-Action Flow Model for General Robot Control”, https://www.physicalintelligence.company/blog/pi0.

**Experimental Designs Or Analyses:**

Experiments were conducted on the LIBERO and MetaWorld MT50 benchmarks, as well as on real-world tasks. The quantitative results demonstrate superior performance compared to several baselines, supporting the paper’s claims. Additionally, the authors performed ablation studies to assess the effectiveness of each variant of the proposed method, further reinforcing their findings.

**Methods And Evaluation Criteria:**

STAR leverages rotation-based gradients in its RaRSQ component and uses an autoregressive transformer to capture causal relationships between skills. The evaluation is conducted on established benchmarks (LIBERO and MetaWorld MT50) and validated with real-world robot manipulation tasks. The metrics focus on success rates, and the results indicate that STAR achieves higher performance than competing methods.

**Other Comments Or Suggestions:**

1. Both eq(8) and eq(11) denote r_d. Please clarify whether this is a typo.
2. Including a video demo for the real-world manipulation tasks would be highly beneficial, as metrics beyond success rate—such as completion time—are also critical.

**Other Strengths And Weaknesses:**

Strengths:
1. This paper addresses a critical and practical problem in robot learning: robot skill abstraction.
2. The proposed method is reasonable and effective, utilizing RaRSQ to prevent codebook collapse and CST to model dependencies between skills.
3. The thorough experiments demonstrate the algorithm’s effectiveness, and real-world experiments further support the paper’s claims.
3. Overall, the paper is well-written and easy to follow.

Weaknesses:
1. My main concern is that this approach primarily builds on existing methods. For example, RaRSQ is essentially a combination of VQ-VAE and a rotation trick, while CST integrates a VLA Transformer with autoregressive prediction. This makes it difficult to pinpoint the paper’s unique contribution.

**Questions For Authors:**

1. What is the unique contribution of STAR compared to prior works in VQ-VAE, rotation-based augmentation, and VLA transformers? It is important to highlight the differences.
2. Can you confirm if the use of r_d in both eq(8) and eq(11) is correct, or if it is a typographical error?
3. Could you provide more details or video on the real-world robot experiments as well as completion time or speed?

**Relation To Broader Scientific Literature:**

The work relates to VQ-VAE-based discrete representation learning, causal modeling, and robot learning.

**Theoretical Claims:**

The paper does not present significant theoretical contributions.

---

> ### Author Rebuttal · Authors · 2025-04-01
>
> We appreciate your thoughtful comments and positive assessment of our work. Below we address the specific questions and concerns:
> ## R1: A broader discussion on recent progress in robot learning—especially work on vision-language-action (VLA) transformers like π0—would provide valuable context regarding performance, efficiency, and scalability.
> Indeed, we have already compared our method with several advanced VLA methods in our manuscript, including OpenVLA (7B) and Octo, as shown in Table. 1. Further, we appreciate and follow your good advice to include π0 in our evaluation. The results on LIBERO benchmark are summarized below:
> | |**Size**|**Pretrain**|**LIBERO-Long**|**LIBERO-Spatial**|**LIBERO-Goal**|**LIBERO-Object**|**Avg**|
> |-|-|-|-|-|-|-|-|
> |π0|3.3B|True|85.2|96.8|95.8|98.8|94.15|
> |Ours|16.2M|False|88.5|95.5|95.0|98.3|94.30|
>
> Despite using significantly fewer parameters (200× smaller) and no pretraining, our approach achieves higher average performance (94.3% vs. 94.15%). Most notably, we observe the largest improvement on LIBERO-Long tasks (+3.3%), which aligns with our method's focus on addressing challenges in long-horizon tasks through effective skill composition.
>
> VLA methods typically follow a pretraining-finetuning paradigm, with performance dependent on pretraining data and architecture (e.g., π0 > OpenVLA). Our work shows lightweight models trained from scratch can match or exceed VLA methods requiring extensive pretraining. Future work could integrate our approach with large language models to further advance robot learning capabilities.
>
> ## Q1: What is the unique contribution of STAR compared to prior works in VQ-VAE, rotation-based augmentation, and VLA transformers?
> We would like to further clarify that our key contributions lie in designing robot-specific residual VQ-VAE that effectively decomposes complex robot behaviors into discrete skills, addressing fundamental challenges in robot skill learning.
> ### (1) Robot-Specific Solution to Codebook Collapse in Skill Learning
> Rather than a straightforward application of rotation tricks to VQ-VAE, RaRSQ is specifically designed for robotic skill space with hierarchical structure. While residual VQ-VAE offers larger representation space, this makes codebook collapse more problematic. Our integration of rotation-based gradients within the residual framework preserves geometric relationships throughout hierarchical action quantization, effectively capturing both coarse primitives (move, pick) and fine-grained adjustments required for complex manipulation.
> ### (2) Explicit Modeling of Hierarchical Skill Dependencies
> Our CST differs fundamentally from previous VLA transformers by explicitly modeling conditional dependencies between different skill abstraction levels. Unlike approaches that predict actions directly or treat skills independently, CST captures structured relationships between coarse and fine-grained behaviors. As shown in Fig. 7, it reveals distinct skill dependency patterns that validate this approach. For example, given a first-level code, we found strong preference for specific second-level codes, showing how coarse skills constrain fine-grained behaviors in ways prior approaches cannot capture.
>
> As shown in Table. 2, removing the autoregressive component significantly reduces performance, , especially in LIBERO-Goal (-6.9%) and LIBERO-Long (-5.2%). In addition, removing both components creates a 5.8% drop over all tasks, exceeding the sum of individual effects, demonstrating the synergistic relationship between skill representation and composition.
> ## Q2: Can you confirm if the use of r_d in both eq(8) and eq(11) is correct, or if it is a typographical error?
> Thanks for pointing out the typo. $r_d$ in eq(11) should be $\hat{r_d}$, and we will correct it in the final revision.
> ## Q3: Could you provide more details or video on the real-world robot experiments as well as completion time or speed?
> We have recorded comprehensive videos showing the complete execution sequences for both tasks (drawer manipulation and sequential object placement). Since ICML rebuttal guidelines explicitly support anonymous links for supplementary figures and tables but don't specifically videos, complete videos will be included on our future project page.
>
> Regarding completion time, STAR demonstrates favorable completion times, averaging 29.6 seconds for the sequential object placement task and 37.7 seconds for the drawer manipulation task. This efficiency is attributable to our hierarchical skill abstraction approach, where a single inference step produces an action chunk (consisting of eight atomic actions in our implementation). This reduces the number of required inference steps and consequently decreases overall execution time. We will incorporate these timing metrics in the final version to provide a more comprehensive evaluation.

---

> > ### Comment · Reviewer_Lv7s · 2025-04-02
> >
> > Thank you for the author’s response. It addresses most of my concerns. Overall, the paper appears to be of good quality, and I am inclined to recommend its acceptance.

---

> > > ### Author Response · Authors · 2025-04-03
> > >
> > > Thank you once again for your recognition of our work and your support for its acceptance. We truly appreciate your valuable feedback.

---

### Official Review · Reviewer_G1Cu · 2025-03-21

**Overall Recommendation:** 4

**Summary:**

The paper proposes to improve prior latent discrete policies by preventing codebook collapse, improving the codebook utilization and proposes to use an autoregressive poiicy to chain together the various discrete skills. To achieve better codebook utilization (and preventing code collapse), the paper proposes to use rotation-augmented residual skill quantization. Essentially, this strategy introduces a hierarchical, coarse-to-fine latent vector codebook where the latents are encoded at different depths. Finally, a causal transformer predicts these latent vectors (again in a coarse to fine fashion) and decodes actions for the downstream policy. The proposed training strategy is tested on LIBERO and Meta-World, and shows superior performance than prior SOTA methods as well as ablative versions of the proposed method which do not use the rotation-augmented codebook strategy and the auto-regressive decoding process.

## update after rebuttal
I have read all reviewer's review and author rebuttal. I think the author responses make sense and address most concerns raised by other reviewers. Hence, I will vote for acceptance.

**Claims And Evidence:**

Yes, in my reading, the claims are: a) that the proposed strategy leads to better use of codebooks (which is shown in section 4.3), and that leads to overall better performance than prior latent variable models (shown in Table-1) and the proposed changes help (shown in ablations of Table-2)

**Essential References Not Discussed:**

No

**Experimental Designs Or Analyses:**

Yes -- the experiments are sound and show that the proposed method is better than SOTA and that the proposed changes help improve performance.

**Methods And Evaluation Criteria:**

Yes -- the proposed method is well motivated and works on improving the codebook utilization, and the benchmark datasets are reasonable (i.e. they are standard) and make sense for the problem.

**Other Comments Or Suggestions:**

N/A

**Other Strengths And Weaknesses:**

In general, the paper is well-written and easy to follow.

**Questions For Authors:**

N/A

**Relation To Broader Scientific Literature:**

The key contribution is to improve the performance of latent variable based policies by designing a strategy to improve codebook utilization and thus finally the downstream performance on manipulation tasks.

**Theoretical Claims:**

N/A

---

> ### Author Rebuttal · Authors · 2025-03-31
>
> We sincerely thank Reviewer G1Cu for the thorough and positive assessment of our work. We appreciate your recognition of our key contributions in improving codebook utilization through rotation-augmented residual skill quantization and implementing autoregressive decoding for effective skill composition.
>
> If you have any further questions, we would be more than happy to address them.

---

> > ### Comment · Reviewer_G1Cu · 2025-04-02
> >
> > I have read all reviewer's review and author rebuttal. I think the author responses make sense and address most concerns raised by other reviewers. Hence, I will vote for acceptance.

---

> > > ### Author Response · Authors · 2025-04-03
> > >
> > > Thank you once again for your recognition of our work and your support for its acceptance. We truly appreciate your valuable feedback.

---

### Decision · Program_Chairs · 2025-05-01

**Decision:**

Accept (spotlight poster)

**Comment:**

The paper proposes a novel VQ-VAE for action sequences that builds a hierarchical coarse-to-fine latent vector codebook where the latents are encoded at different depths, by considering rotation-augmented residual skill quantization. The paper motivates the proposed method  as a way to mitigate mode collapse in vanilla VQ-VAEs. A causal transformer predicts the sequence of latents per skill (action sequence). They show in LIBERO improved results over alternatives. The paper addresses an important problem of discretizing action sequences, and proposes a working solution.  All reviewers recommend acceptance.